# The Association between Antibiotic Use and the Incidence of Heart Failure: A Retrospective Case-Control Study of 162,188 Outpatients

**DOI:** 10.3390/biomedicines11020260

**Published:** 2023-01-18

**Authors:** Sven H. Loosen, Sarah Krieg, Julia Gaensbacher, Corinna Doege, Andreas Krieg, Tom Luedde, Mark Luedde, Christoph Roderburg, Karel Kostev

**Affiliations:** 1Department of Gastroenterology, Hepatology and Infectious Diseases, University Hospital Duesseldorf, Medical Faculty of Heinrich Heine University Duesseldorf, 40225 Duesseldorf, Germany; 2Internal Medicine III, University of Schleswig Holstein, Campus Kiel, 24105 Kiel, Germany; 3Department of Pediatric Neurology, Center of Pediatrics and Adolescent Medicine, Central Hospital Bremen, 28211 Bremen, Germany; 4Department of Surgery (A), University Hospital Duesseldorf, Medical Faculty of Heinrich Heine University Duesseldorf, 40225 Duesseldorf, Germany; 5KGP Bremerhaven, 27574 Bremerhaven, Germany; 6Epidemiology, IQVIA, 60549 Frankfurt, Germany

**Keywords:** antibiotics, heart failure, inflammation, intestinal microbiome, gut microbiota

## Abstract

The pathogenesis of heart failure (HF) is multifactorial, and is characterized by structural, cellular, and molecular remodeling processes. Inflammatory signaling pathways may play a particularly understudied role in HF. Recent data suggest a possible impact of antibiotic use on HF risk. Therefore, the aim of this retrospective case-control study was to investigate the association between antibiotic use and the incidence of HF. Data from the Disease Analyzer (IQVIA) database for patients diagnosed with HF and matched non-HF controls from 983 general practices in Germany between 2000 and 2019 were analyzed. A multivariable conditional logistic regression model was performed. Regression models were calculated for all patients, as well as for data stratified for sex and four age groups. A total of 81,094 patients with HF and 81,094 patients without HF were included in the analyses. In the regression analysis, low, but not high, total antibiotic use was significantly associated with a slightly lower HF risk compared with non-antibiotic users (OR: 0.87; 95% CI: 0.85–0.90). A significantly lower HF incidence was observed for sulfonamides and trimethoprim (OR: 0.87, 95% CI: 0.81–0.93) and for macrolides (OR: 0.87, 95% CI: 0.84–0.91). High use of cephalosporins, however, was associated with an increased HF risk (OR: 1.16; 95% CI: 1.11–1.22). In conclusion, this study from a large real-world cohort from Germany provides evidence that the use of different antibiotics may be associated with HF risk in a dose-dependent manner, possibly due to involved inflammatory processes. Overall, this study should provide a basis for future research to offer new therapeutic strategies for HF patients to improve their limited prognosis.

## 1. Introduction

Heart failure (HF) represents one of the most common causes of disease and reasons for hospitalization in industrialized nations [1,2]. It is a clinical syndrome characterized by cardinal symptoms (e.g., dyspnea, ankle edema, and fatigue) that may be accompanied by specific clinical examination findings (e.g., elevated jugular venous pressure, pulmonary venous congestion, and peripheral edema). Cardiac dysfunction is based on cardiac structural defects and/or dysfunction resulting in elevated intracardiac pressures, and/or inadequate cardiac output at rest and/or during exercise. Etiologically, there may be underlying myocardial dysfunction (systolic, diastolic, or both) or diseases of the heart valves, pericardium, and endocardium, as well as cardiac rhythm and conduction disorders [3,4]. In Europe, its prevalence is 1–2% [4], rising to more than 10% in men and 8% in women over 70 years of age [3]. Despite numerous therapeutic options, the mortality rate of patients with HF remains still high [5]. Several mechanisms appear to be involved in HF pathogenesis. Contributing factors include increased hemodynamic overload, ischemia-related dysfunction, and ventricular remodeling [3,4]. Although understudied to date, inflammatory signaling pathways may play another important role in the development and progression of HF [6,7]. In this context, the intestinal microbiome has recently come into focus as a factor that may influence HF [8,9]. Interestingly, a negative effect of antibiotic use on the intestinal microbiome has already been demonstrated in several disease entities [10]. However, determining whether antibiotics have a potential impact on the progression or development of HF has not yet been addressed in detail.

Given the lack of similar data so far, the aim of our study was to investigate the association between antibiotic use and HF incidence in a large outpatient population in Germany. For this purpose, different antibiotic classes and doses were considered, with patients stratified into non-users, and low (up to the 50th percentile) and high (above the 50th percentile) users of antibiotics, both overall and for each antibiotic class.

## 2. Materials and Methods

### 2.1. Database

This study was based on data from the Disease Analyzer database (IQVIA). This database includes anonymous demographic data, diagnoses, and prescriptions. The coverage is approximately 3–5% of all general practices (GP) in Germany. The sampling method for the Disease Analyzer database uses summary statistics from all doctors in Germany, published yearly by the German Medical Association. IQVIA determines the panel design according to the German federal state, specialist group, community size category, and age of physician [11]. 

### 2.2. Study Population

This retrospective case-control study included adult patients (≥18 years) with an initial HF diagnosis (ICD-10: I50) in 983 GP in Germany between January 2010 and December 2019 (index date; Figure 1). A further inclusion criterium was an observation time of at least five years prior to the index to evaluate the duration of antibiotic use. HF cases were matched to non-HF controls by sex, age, and pre-defined chronic co-diagnoses documented within five years prior to the index date (obesity, diabetes, hypertension, ischemic heart diseases, cancer, and chronic obstructive lung disease). For the controls, the index date was that of a randomly selected visit between January 2010 and December 2019 (Figure 1). 

### 2.3. Study Outcomes 

The main outcome of the study was the association between the incidence of HF diagnosis and prescriptions of antibiotic drugs in total, as well as tetracyclines, penicillins, cephalosporins, sulfonamides and trimethoprim, macrolides, and quinolones separately. For each patient, the total count of prescribed milligrams (mg) within five years prior to the index date was calculated. Patients were categorized as non-users, low (up to 50th percentile) or high (above 50th percentile) users of antibiotics, both in total and for each antibiotic class, based on the total sum of prescribed mg within five years prior to the index date. 

### 2.4. Statistical Analyses

Differences in the sample characteristics between those with and those without HF were tested using chi-squared tests for categorical variables and Wilcoxon tests for age. A multivariable conditional logistic regression model was conducted to study the association between antibiotic use (yes/no), antibiotic amount (low and high users vs. non-users) and HF. Regression models were calculated for all patients, as well as stratified by sex and four age groups (age 18–60, age 61–70, age 71–80, age > 80). A *p*-value of <0.001 was considered to be statistically significant as large patient samples were analyzed and multiple comparisons were conducted. Analyses were carried out using SAS version 9.4 (SAS Institute, Cary, NC, USA).

## 3. Results

### 3.1. Basic Characteristics of the Study Sample

The present study included 81,094 HF patients and 81,094 non-HF controls. The basic characteristics of the study patients are displayed in Table 1. Mean age (SD) was 74.1 (11.9) years; 51.4% of patients were female. There were no differences in the prevalence of selected co-diagnoses between cases and controls.

### 3.2. Association of Antibiotic Use with Heart Failure

In regression analysis, low, but not high, use of antibiotics in total was significantly associated with a slightly decreased rate of HF (OR: 0.87; 95% CI: 0.85–0.90) compared to non-users. Interestingly, a significantly decreased rate of HF was observed for low use of sulfonamides and trimethoprim (OR: 0.87, 95% CI: 0.81–0.93), as well as for macrolides (OR: 0.87, 95% CI: 0.84–0.91). In contrast, high use of cephalosporins was associated with an increased rate of HF (OR: 1.16; 95% CI: 1.11–1.22; Table 2).

### 3.3. Age- and Sex Stratified Analyses

The negative association between HF and low antibiotic use in total, as well as low use of macrolides, was observed in both women and men, as well as in three age groups (≤60, 61–70, 71–80 years; Table 3). The positive association between cephalosporins and HF was observed in women only, as well as in two age groups (71–80, >80 years). 

## 4. Discussion

Using the population-based Disease Analyzer Database (IQVIA), this was the first study to assess a large real-world population from Germany totaling more than 81,000 adult outpatients with and without HF in relation to antibiotic use. The data showed that, compared with non-use of antibiotics, only low-dose but not high-dose antibiotic use was associated overall with a slightly decreased rate of HF. This effect was observed primarily for sulfonamides and trimethoprim, as well as for macrolides. In contrast, high-dose use of cephalosporins was associated with an increased risk of HF. However, in this study, no clear cross-substance association between antibiotic use and increased or decreased incidence of HF was found. 

The pathogenesis of HF is known to be multifactorial, and is characterized by structural, cellular, and molecular remodeling processes involving numerous inflammatory and metabolic mechanisms [3]. Interestingly, already by 1999, Niebauer et al. hypothesized that HF might be influenced by the use of antibiotics. In patients with advanced HF, the authors detected increased levels of endotoxin, also known as lipopolysaccharide (LPS), in the blood, with levels doubling when edema was present and decreasing again when edema was treated with diuretics [12]. Recently, attention has focused on the intestinal microbiome and the “heart-gut axis” as potential factors influencing the development and progression of HF [8,13]. Studies have shown that patients with HF have significantly reduced diversity of the gut microbiota and downregulation of major intestinal bacterial groups compared with healthy individuals [8,14,15]. Thus, the gut microbiota itself could promote systemic inflammatory processes, although it is unclear whether altered gut microbiota is a cause of HF or a secondary effect of the disease [8]. Interestingly, it is suggested that the intestinal microbiome is significantly affected by antibiotic use [15]. There is evidence of changes in the intestinal microbiome in studies showing that selective decontamination of the digestive tract with antibiotics decreases gut LPS content, expression of the monocyte LPS co-receptor CD14, intracellular monocyte production of interleukin-1β (IL-1β), interleukin-6 (IL-6), and tumor necrosis factor (TNF)-α [16]. Manipulation of the composition, distribution, and potential functionality of the intestinal microbiome by antibiotic use might alter the inflammatory and metabolic environment affecting the cardiovascular system and cardiac remodeling [17]. Interestingly, a small pilot study of patients with HF and reduced ejection fraction (EF) showed that treatment with the probiotic yeast *Saccharomyces boulardii* (*S.boulardii*) resulted in a greater increase in the left ventricular ejection fraction (LVEF) compared with a placebo [18]. Similarly, the multicenter, prospective, randomized, open-label Targeting Gut Microbiota to Treat Heart Failure (GutHeart) trial investigated a treatment benefit of additional administration of *S.boulardii* or the oral non-absorbable antibiotic rifaximin in HF patients with reduced ejection fraction, despite optimal drug therapy in an intention-to-treat analysis. However, in contrast to the observations of Constanza et al., three months of treatment with *S.boulardii* or alternatively with rifaximin, in addition to guideline-directed HF therapy, had no significant effect on LVEF, microbiota diversity, or measured biomarkers [9]. 

In HF patients, reduced cardiac output and congestion are known to lead to ischemia and edema in the intestinal wall. Structural and functional changes increase intestinal permeability and induce secondary inflammation [19]. Of note, studies have shown that the severity of HF symptoms correlates with the extent of intestinal permeability, the amount of pathogenic bacteria, and secondary inflammation [14]. In this context, the leakage of bacterial products such as LPS across the gut barrier [14,20] is thought to trigger a systemic inflammatory response, including activation of Toll-like receptor 4 (TLR4) on cells of the innate immune system [21]. The release of pro-inflammatory cytokines, such as TNF-α, is in turn reported to lead to a reduction in mitochondrial activity, altered calcium homeostasis, and impaired β-adrenergic signaling in cardiomyocytes [7]. Similarly, other pro-inflammatory cytokines, such as IL-1 and IL-6, have been linked to myocardial dysfunction [7]. Furthermore, as part of the inflammatory process, the gut microbe-generated metabolite trimethylamine-N-oxide (TMAO) has been associated with increased risk of cardiovascular events and the development, severity, and prognosis of HF [22,23,24].

However, the question of why the different antibiotic classes showed different effects on HF risk in this study still remains open. Since the indication for using the broad-spectrum antibiotic cephalosporins class, which are mainly used for pulmonary and urinary tract as well as soft tissue infections [25], corresponds to a spectrum similar to that of macrolides, we consider an association between the antibiotic prescription indication and the diagnosis of HF to be less likely. Otherwise, it could be assumed that a secondary cause such as pneumonia was treated with antibiotics during cardiac decompensation, which would lead to an inverse association between antibiotic use and diagnosis of HF. In contrast, the antibiotics sulfonamides and trimethoprim, which resulted in a reduced risk of HF at low doses in our study, have a narrower therapeutic spectrum, being an option for first-line treatment of acute uncomplicated cystitis [26]. An alternative hypothesis that could explain the divergent effect of different antibiotic classes on HF risk is that the antibiotic classes may affect the gut microbiota differently. In this context, for example, it is known that a rich and diverse intestinal microbiota prevents *Clostridium difficile* infections (CDI), whereas disruption of the intestinal microbiota due to antibiotic use may in turn lead to loss of colonization resistance and proliferation of *C. difficile* [27]. Indeed, antibiotic use represents the most important risk factor for CDI [28,29]. Previous studies analyzed the different antibiotic classes implicated in the greatest risk for CDI. Interestingly, cephalosphorins are among the classes with the highest risk of CDI, whereas the use of macrolides and trimethoprim was associated with a lower risk of CDI [27,30].

Our study also provides evidence that different antibiotic amounts may have a differential impact on HF risk. Accordingly, appropriate clinical use of antibiotics to treat or prevent HF may be significant. Further studies are needed to conclusively address this question, particularly at the molecular pathological and cellular levels, such as examining the effects of different antibiotic classes and doses on cardiomyocytes.

Surprisingly, our data found no evidence of an increased HF risk with the use of fluoroquinolones. In 2019, the German Federal Institute for Drugs and Medical Devices (BfArM) and the European Medicines Agency (EMA) published a red-hand letter providing information about the risk of heart valve regurgitation and/or insufficiency associated with the use of fluoroquinolones. The risk of mitral and aortic valve regurgitation or insufficiency is twice as high in patients taking systemic fluoroquinolones compared to those taking other antibiotics (amoxicillin or azithromycin) [31]. In this context, an experimental study demonstrated that exposure to the fluoroquinolone ciprofloxacin resulted in collagen degradation in aortic myofibroblast cells (13). A possible explanation for the lack of evidence of any effect of fluoroquinolones on HF risk in our study could be that additional predisposing factors, such as congenital or preexisting valvular heart disease, connective tissue diseases such as Marfan syndrome or Ehlers-Danlos syndrome, Turner syndrome, Behçet disease, rheumatoid arthritis, and infective endocarditis usually favor the risk for these fluoroquinolone complications [31]. However, because of their rarity, these conditions may not be represented in the database used for this study.

Our study is subject to limitations, most of which are due to the chosen study design and therefore cannot be avoided [11,32,33]. First, the diagnoses in our database were coded as ICD-10 codes. Therefore, it cannot be excluded that certain diagnoses were misclassified or that data for certain patients were incompletely collected. Second, no information about HF type and stage was available. Third, a selection bias must be assumed, as patients who took antibiotics more frequently presumably had more frequent contact with their physicians than patients who did not take antibiotics. Furthermore, the Disease Analyzer database does not contain data on socioeconomic status (e.g., patients’ education and income) and lifestyle-related risk factors (e.g., smoking, alcohol consumption, and physical activity), so these could not be included in our study. Finally, the present study only allows associations, but no causal relationships can be concluded. Nevertheless, as a strength of this study, we emphasize the large sample size with the use of data from a large real-world patient sample. The IQVIA Disease Analyzer database used for the present analyses was proven to be representative and valid [11].

## 5. Conclusions

This is the first study using a large, real-world population from Germany to investigate the association between HF risk and antibiotic use. Our data provide evidence that the use of different antibiotics might be associated with HF risk in a dose-dependent manner. In this context, inflammatory processes and the intestinal microbiome may play a potential role in the effect of antibiotic use on HF risk. This study should stimulate future efforts to provide further evidence of this association, with the aims of offering new therapeutic strategies to patients with HF and improving their overall limited prognosis. For instance, it would be of interest to investigate the effects of different antibiotic classes and doses on cardiomyocytes at the molecular level.

## Figures and Tables

**Figure 1 biomedicines-11-00260-f001:**
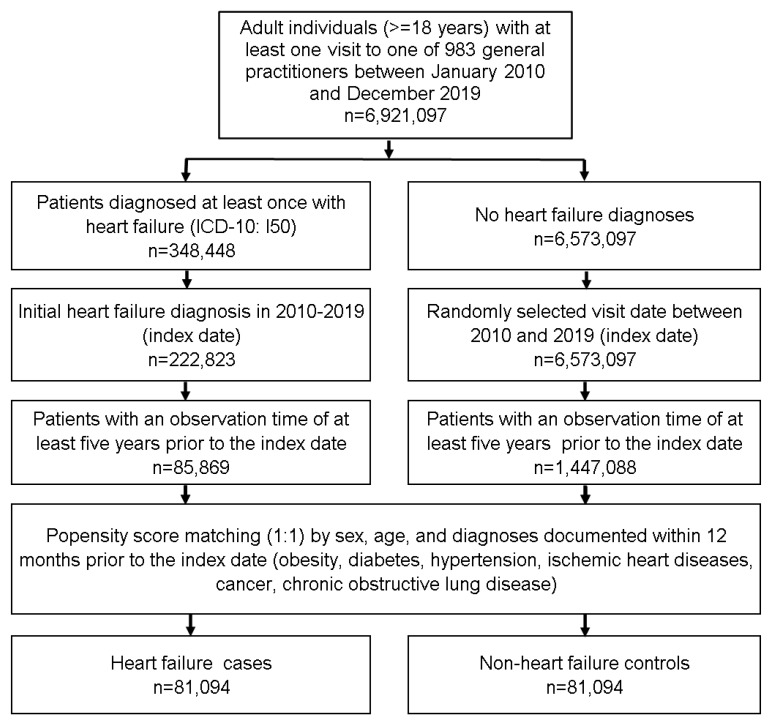
Selection of study patients.

**Table 1 biomedicines-11-00260-t001:** Baseline characteristics of the study sample after 1:1 propensity score matching.

Variable	Proportion Affected among Patients with Heart Failure (%) N = 81,094	Proportion Affected among Patients without Heart Failure (%) N = 81,094	*p*-Value
Age (Mean, SD)	74.1 (11.9)	74.1 (11.9)	1.000
Age ≤ 60	14.0	14.0	1.000
Age 61–70	17.8	17.8	
Age 71–80	34.8	34.8	
Age > 80	33.4	33.4	
Women	51.4	51.4	1.000
Men	48.6	48.6	
Diabetes	31.2	31.2	1.000
Obesity	10.0	10.0	1.000
Hypertension	63.1	63.1	1.000
Ischemic heart diseases	28.3	28.3	1.000
Cancer	12.3	12.3	1.000
Chronic obstructive lung disease	15.5	15.5	1.000
Diseases of esophagus, stomach and duodenum	23.5	23.4	0.456

Proportions of patients are given in %, unless otherwise indicated. SD: standard deviation.

**Table 2 biomedicines-11-00260-t002:** Association between antibiotic therapy and heart failure (multivariable logistic regression models).

Antibiotic Therapy	Proportion among Patients with Heart Failure in %	Proportion among Patients without Heart Failure in %	OR (95% CI)	*p*-Value
Any antibiotic drug				
No therapy	64.0	62.9	Reference	
≤6000 mg	12.7	14.3	0.87 (0.85–0.90)	<0.001
>6000 mg	23.3	22.9	1.00 (0.97–1.02)	0.797
Tetracyclines (J01A)				
No therapy	94.7	94.5	Reference	
≤2000 mg	3.5	3.6	0.98 (0.93–1.03)	0.371
>2000 mg	1.8	1.9	0.93 (0.87–1.00)	0.056
Penicillins (J01C)				
No therapy	88.0	88.3	Reference	
≤20,000 mg	8.3	8.3	1.02 (0.98–1.06)	0.280
>20,000 mg	3.7	3.4	1.05 (1.00–1.11)	0.058
Cephalosporins (J01D)				
No therapy	88.4	89.0	Reference	
≤6000 mg	6.3	6.3	1.02 (0.98–1.06)	0.394
>6000 mg	5.3	4.7	1.16 (1.11–1.22)	<0.001
Sulfonamides and trimethoprim (J01E)				
No therapy	97.1	96.7	Reference	
≤9600 mg	1.8	2.1	0.87 (0.81–0.93)	<0.001
>9600 mg	1.1	1.2	0.94 (0.86–1.03)	0.188
Macrolides (J01F)				
No therapy	87.5	86.6	Reference	
≤3000 mg	6.4	7.2	0.87 (0.84–0.91)	<0.001
>3000 mg	6.1	6.2	0.96 (0.92–1.00)	0.045
Quinolones (J01M)				
No therapy	84.6	84.4	Reference	
≤5000 mg	8.7	9.2	0.96 (0.93–0.99)	0.017
>5000 mg	6.7	6.4	1.04 (1.00–1.09)	0.055

**Table 3 biomedicines-11-00260-t003:** Association between antibiotic therapy and heart failure by sex and age groups (multivariable logistic regression models).

	OR (95% CI)					
Antibiotic Therapy	Women (n = 83,432)	Men (n = 78,756)	Age ≤ 60 (n = 23,066)	Age 61–70 (n = 29,094)	Age 71–80 (n = 56,500)	Age > 80 (n = 53,528)
Any antibiotic drug						
≤6000 mg	0.86 (0.83–0.90) *	0.88 (0.84–0.92) *	0.75 (0.70–0.81) *	0.81 (0.76–0.87) *	0.87 (0.83–0.91) *	0.98 (0.93–1.03)
>6000 mg	0.99 (0.96–1.02)	1.01 (0.97–1.04)	0.90 (0.85–0.95) *	0.94 (0.89–1.00)	1.04 (0.99–1.08)	1.05 (1.01–1.10)
Tetracyclines (J01A)						
≤2000 mg	0.97 (0.90–1.05)	0.98 (0.91–1.06)	0.89 (0.79–1.01)	0.88 (0.78–0.98)	1.00 (0.91–1.10)	1.15 (1.03–1.28)
>2000 mg	0.97 (0.87–1.08)	0.90 (0.81–0.99)	0.95 (0.81–1.11)	0.87 (0.75–1.01)	0.97 (0.86–1.10)	0.95 (0.80–1.13)
Penicillins (J01C)						
≤20,000 mg	1.02 (0.97–1.07)	1.03 (0.97–1.08)	1.01 (0.93–1.10)	0.98 (0.90–1.07)	1.01 (0.94–1.07)	1.08 (1.01–1.15)
>20,000 mg	1.02 (0.95–1.11)	1.08 (1.01–1.17)	0.97 (0.87–1.08)	1.07 (0.95–1.20)	1.15 (1.04–1.26)	1.05 (0.93–1.17)
Cephalosporins (J01D)						
≤6000 mg	1.03 (0.97–1.09)	1.01 (0.95–1.07)	0.98 (0.89–1.09)	1.05 (0.96–1.16)	1.03 (0.95–1.10)	1.01 (0.94–1.09)
>6000 mg	1.23 (1.15–1.31) *	1.10 (1.03–1.17)	1.18 (1.07–1.31)	1.16 (1.04–1.28)	1.15 (1.06–1.25) *	1.17 (1.07–1.28) *
Sulfonamides and trimethoprim (J01E)						
≤9600 mg	0.88 (0,81–0.95)	0.81 (0.69–0.95)	0.77 (0.63–0.96)	0.81 (0.68–0.98)	0.85 (0,75–0,96)	0.92 (0.82–1.03)
>9600 mg	0.90 (0.81–1.00)	1.07 (0.89–1.29)	1.05 (0.81–1.36)	1.12 (0.89–1.42)	0.93 (0.79–1.09)	0.83 (0.71–0.97)
Macrolides (J01F)						
≤3000 mg	0.89 (0.84–0.94) *	0.86 (0.81–0.91) *	0.82 (0.76–0.90) *	0.81 (0.74–0.88) *	0.88 (0,82–0.94) *	0.98 (0.91–1.06)
>3000 mg	0.97 (0.91–1.02)	0.95 (0.90–1.01)	0.93 (0.85–1.02)	0.90 (0.82–0.99)	1.00 (0.93–1.08)	1.00 (0.92–1.09)
Quinolones (J01M)						
≤5000 mg	0.95 (0.90–0.99)	0.97 (0.92–1.03)	0.83 (0.75–0.91) *	0.97 (0.89–1.06)	1.00 (0.94–1.06)	0.95 (0.90–1.01)
>5000 mg	1.03 (0.97–1.09)	1.06 (0.99–1.13)	1.02 (0.92–1.13)	1.02 (0.93–1.12)	1.07 (1.00–1.15)	1.03 (0.95–1.11)

* *p* < 0.001.

## Data Availability

The datasets used and analyzed during the current study are available from the corresponding author on reasonable request.

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
