# Peer review of "The Association between Antibiotic Use and the Incidence of Heart Failure: A Retrospective Case-Control Study of 162,188 Outpatients"

_biomedicines, 2023, doi:10.3390/biomedicines11020260_

Round 1

Reviewer 1 Report

The manuscript by Sven Loosen et al., submitted to Biomedicines is an interesting overview of the connection between the antibiotics use and HF. The logistic regression model is well designed, and the data is well organized with a wealth of information presented. However, some adjustments need to be taken into consideration by the authors.

1. In Table 1, what evidences could the authors provide to demonstrate that no differences in the prevalence of selected co-diagnoses between two groups?

2. “Heat-gut axis” is known as to induce the development of HF, but the reverse is also the fact. Why don’t you consider the gastrointestinal dysfunction as the co-diagnosis of HF in this context?

3. Maybe the question that different antibiotic amount caused a different influence on HF risk in this study should be discussed. How do the authors consider the issue that an appropriate clinical use of antibiotics to treat or prevent HF?

Reviewer 2 Report

Biomedicines-2150115, The association between antibiotics use and the incidence of heart failure: a retrospective case-control study of 162.188 outpatients by Sven H Loosen et al. The authors aimed in their study to investigate the association between antibiotic use and HF incidence in a large outpatient population in Germany.

Comments

Abstract 

- The aim of the study is not stated clearly. 

- The conclusion needs to highlight more the importance of the study findings.

Introduction

-  Page 1, lines 39-40: The reviewer suggests defining “Heart failure” more specifically.

- Page 2, lines 53-54: “Therefore, different classes of 53 antibiotics should be assessed and patients should be categorized as non-users.”. The verb tense needs to be changed to be in the past. Accordingly, please replace “should be” with “were”.

 - Page 2, lines 54-55: It is not clear how the authors categorized the patients as non-users, low (up to 50th percentile) and high (above 50th percentile) users. What is your scale? the five years prior to the index date or the amount of antibiotic? Check the writing style “50th and 50th" and make it constant throughout the sentence/manuscript.

Materials and Methods 

-Page 2, line 62: The reviewer suggests removing the reference since there is no need for it at this place. 

- Page 2, lines 66-67: “In Germany, the sampling … and specialized practices”. What do the authors want to refer to?

- Page 2, lines 69 & 71: The reference style needs to be corrected to match the Journal’s style.  Also, please check this reference “Roderburg et al., 2021” as it is not in the reference list. Moreover, “Loosen et al., 2022” is not studying HF as it is referred to in the text.

- Page 3, Figure 1: What do the authors mean by “At least one heart failure diagnosis”??

- For the HF group, it is not clear whether the patients diagnosed with HF before or during the use of the antibiotics? What is the type of HF (ischemic/non-ischemic) and the stage (ACC/AHA or NYHA (if possible)? Since the manuscript focus is HF, the authors should provide more details about it (type and stage; whenever possible).

Results

-Tables 2 & 3: In the title the authors mentioned that they used the “multivariable logistic regression models” while in “Abstract” and “Statistical Analysis”, they mentioned using the “univariable conditional logistic regression model”, could you please clarify? 

Discussion

- The reviewer suggests adding a diagram illustrating the proposed mechanism of the inflammatory signaling pathways in the development and progression of HF.

Round 2

Reviewer 2 Report

The authors have done a nice job of revising this manuscript and addressing the reviewer comments. The manuscript now reads with greater focus and clarity. I have two additional minor comments for the authors to consider:  

1- Page 3, Figure 1: The reviewer suggests replacing “At least one heart failure diagnosis” with “Diagnosed at least once with HF” to sound more scientific.

2- Materials and methods: I would still suggest adding some details about the type of HF and stage per my previous comment in the first round of review; whenever possible.

Author Response

Response: Thank you for your valuable comments. We will of course address your other comments point by point in the following.

  1. Page 3, Figure 1: The reviewer suggests replacing “At least one heart failure diagnosis” with “Diagnosed at least once with HF” to sound more scientific.

Response: Figure 1 was replaced.

  1. Materials and methods: I would still suggest adding some details about the type of HF and stage per my previous comment in the first round of review; whenever possible.

Response: Database used does not contain information on type and stage of HF what is why we added this into limitations.
